# A Symbolic Encapsulation Point as Tool for 5G Wideband Channel Cross-Layer Modeling

**DOI:** 10.3390/e22101151

**Published:** 2020-10-14

**Authors:** Nenad Stefanovic, Marija Blagojevic, Ivan Pokrajac, Marian Greconici, Yigang Cen, Vladimir Mladenovic

**Affiliations:** 1Faculty of Technical Sciences Cacak, University of Kragujevac, 34000 Kragujevac, Serbia; snomb@mts.rs (N.S.); marija.blagojevic@ftn.kg.ac.rs (M.B.); 2Electronic Systems Department, Military Technical Institute, 11000 Belgrade, Serbia; ivan.pokrajac@vs.rs; 3Fundamental of Physics for Engineers Department, Politehnica University Timisoara, 300223 Timisoara, Romania; marian.greconici@upt.ro; 4School of Computer and Information Technology, Beijing Jiaotong University, Beijing 100044, China; ygcen@bjtu.edu.cn

**Keywords:** LCR, AFD, 5G channel model, FSMC, symbolic encapsulation point, cross-layer

## Abstract

Considering that networks based on New Radio (NR) technology are oriented to provide services of desired quality (QoS), it becomes questionable how to model and predict targeted QoS values, especially if the physical channel is dynamically changing. In order to overcome mobility issues, we aim to support the evaluation of second-order statistics of signal, namely level-crossing rate (LCR) and average fade duration (AFD) that is missing in general channel 5G models. Presenting results from our symbolic encapsulation point 5G (SEP5G) additional tool, we fill this gap and motivate further extensions on current general channel 5G. As a matter of contribution, we clearly propose: (i) anadditional tool for encapsulating different mobile 5G modeling approaches; (ii) extended, wideband, LCR, and AFD evaluation for optimal radio resource allocation modeling; and (iii) lower computational complexity and simulation time regarding analytical expression simulations in related scenario-specific 5G channel models. Using our deterministic channel model for selected scenarios and comparing it with stochastic models, we show steps towards higherlevel finite state Markov chain (FSMC) modeling, where mentioned QoS parameters become more feasible, placing symbolic encapsulation at the center of cross-layer design. Furthermore, we generate values within a specified 5G passband, indicating how it can be used for provisioningoptimal radio resource allocation.

## 1. Introduction

Starting from 5G NR access networks, the customization of particular radio links for a variety of services, traffic patterns, end-user demographics, and specific user experience of data [1], brings quality of service (QoS) metrics as prior demands, promoting 5G as the first QoS-driven radio network [2]. Apart from previous networks, based on a general purpose framework (4G, 3G), the 5G NR approach is a customized, user-optimized concept for each use case and scenario, where particular throughput, latency, packet error rate, packet delay budget, or some other QoS metric is targeted [3]. Furthermore, time-varying channel properties become more significant, especially in vehicle-to-vehicle (V2V) and other similar mobile scenarios. This leads to a key difference of 5G NR channels with respect to conventional cellular systems—non-stationarity [4]. To investigate data transfer quality, in these circumstances, channel modeling is indispensable. In static channels, first-order statistics such as probability density function (PDF) and cumulative distributive function (CDF) are sufficient for modeling and performance prediction. Usually, they follow one of the well-known multipath fading first-order statistics—Rice, Rayleigh, Weibull, Nakagami, and others, allowing reliably modeling, predicting, and calculating of targeted performance metrics. In non-stationary channels, as PDF and CDF are time-varying, analysis must be complemented with higher order statistics, following the path to performance measurement or evaluation. For most applications, second-order statistics, such as LCR and AFD (average fade), represent enough statistical metrics to express this non-stationarity [5,6] and non-linearity feature [7]. We put our motivation point to the fact that representative general channel 5G modeling tools lack in second-order statistics analysis, although it may represent crucial metadata between physical channel parameters (velocity, direction angles, etc.) and information metrics (energy efficiency, packet error rate, and others)—cross-level design. This conclusion comes from the fact that only one of the top-ten general channel models, according to the acknowledged reference, is able to support both dual-mobility and high-mobility, as two of the four most challenging scenarios in 5G. On the other hand, observations made in scenario-specific 5G channel models of mobile channels with LCR and AFD evaluation, however, do not provide generality in terms of supporting various 5G technologies and adapting to different scenarios, as general channel 5G models do. Second, all observations are made in the narrowband sense, limiting perception for band part resource scheduling. Third, analytical expressions for LCR and AFD are often complex, comprising integral functions that diverge [8] and are consequently, computationally intensive.

Assuming some typical 5G propagation scenarios (Figure 1), as for contributions of this work, we bring: (i) an additional tool for encapsulating different mobile general channel 5G modeling approaches; (ii) wideband extension of LCR and AFD evaluation for optimal radio resource allocation modeling; and (iii) lower computational complexity and simulation time reduction related to scenario-specific 5G channel models.

We begin with Clarke’s deterministic model for the narrowband case, as adopted in [9] and then, extend it to the wideband case. The assumptions considered include: vertically polarized signal, point scatterers on the horizontal plane, and a ray tracing approach to compute the distance matrix and signal’s complex envelope in each sample point of the mobile route. We discard eventual comments on model’s accuracy by keeping a focus on the means to capture the signal’s complex envelope and process its statistics, as an object of contribution. However, a scenario that considers multiplane propagation and objects with larger radar cross-sections could be analyzed also, by introducing the vertical axis coordinate (scatterer’s height) in distance matrix computation and several point scatterers at a single position. Observations are made on small-scale routes (small area) as common ground when investigating fast fading behavior. After retrieving statistics from the sampled complex envelope, we proceed to define appropriate channel states and derivea probability transition matrix as building blocks of the two-state FSMC model. That way, we enter into a more comfortable, information theory environment to estimate QoS performance [10].

Complete observation continues in the direction of unifying the models into one environment. This provides results in one place with the properties to simplify all complex calculations in two directions. The first direction is to provide all observations in closed-form expressions. Such expressions are further simplified and manipulated, and this provides a better insight into the observed problem that is being solved. The second direction is to obtain numerical values that give all the characteristics of the process or system. These results are shown in one place.

The rest of the paper is organized as follows: in Section 2, we give a short review of related work on second-order statistics application and analysis and achievements in 5G channel modeling, relying on two robust surveys. Section 3 provides the roadmap from the viewpoint of encapsulation and integration of symbolic and numerical analysis. In Section 4, starting from Clarke’s well-known deterministic model in the Rayleigh channel, we describe a modeling tool for small areas and steps to capture a complex envelope and evaluate its statistics. Hereafter, we continue with FSMC formulation, correlated to these statistics and in Section 5, we put our simulation setup observing three specific 5G scenarios, transmission schemes, and corresponding measurements in physical resource blocks (PRB). Finally, in Section 6, we take some conclusions on the use of second-order statistics in 5G networks as metadata for higher layer modeling and optimal radio resource allocation.

## 2. Related Work

We observe two sides of applications following the topic of LCR and AFD. One side considers relations to the higher network levels where, according to comprehensive survey [11], applications of LCR and AFD are used through Markov chain and decision process synthesis, handoff algorithms analysis to packet error rate (PER) evaluation [12], packet buffer length optimization, adaptive modulation and coding with interleaved and non-interleaved schemes creation, and throughput estimation for different communication protocols, with given reference to papers covering the matter. On the other side, many investigations are done to estimate these signal statistics regarding specific channel configurations [13,14,15] and various signal distributions [16].

Channel modeling for 5G has evolved during the past few years. Much effort was shown in the survey in [17] to aggregate the related topic. Related authors distinguish three approaches to the field: channel measurements, scenario-specific 5G channel modeling, and general 5G models. Only few measurements and scenario-specific channel models tackled LCR and AFD in order to characterize various aspects of the dynamic behavior of envelope fluctuations in [18], and use it for higher layer (FSMC) modeling scenarios [19,20]. Although in [19], wideband was observed, neither of them included these metrics in the wideband aspect. In [21], the validity of the FSMC approach was confirmed by field measurements in a case-specific, high-speed train (HST) scenario, but there is not found in previous papers for the general 5G model. Furthermore, the complexity of analytical expressions derived in these scenario-specific models is often numerically intensive. From the general channel 5G modeling view, the survey also includes 3GPP’s study of actual channel model tools for next generation network evaluation and design [22]. First comes 3GPP’s NR (New Radio) channel model in the form of tapped delay line (TDL) and clustered delay line (CDL) channels and implemented in MATLAB’s 5G Toolbox, bringing all relevant setups for testing, signal generation and analysis, as map-based propagation simulating scenarios [23]. In QuaDRiGa (QUAsi Deterministic RadIo channel GenerAtor) [24], a geometry-based stochastic channel model is used of multiple-input multiple-output (MIMO) radio links, providing all relevant parameters as absolute path delays, Doppler spread, channel impulse response (CIR), etc. Based on this model, Quamcom proposed a simulator [25] for further research and development of the 5G NR physical layer [26]. Statistical, cluster-based, modeling tool, COST 2100 [27], was developed and used for analysis, some time before ITU-R’s vision of 2020 and beyond, focused on multi-user, distributed MIMO and moving Tx-Rx scenarios in the 4G communication system. The program METIS [28] belongs to the deterministic types of channel modeling tools and was intended to complement time-consuming and expensive measurement campaigns for overall 5G scenarios, offering a wide range of parameters and scenario’s setup, but, as previous, finalizing just with PDF and CDF statistics. To mention at the end, but not as final on the list, the NYUSIM model [29], built up from extensive propagation measurements carried out during a time period of four years and proposed as a statistical spatial channel model using a time cluster–spatial lobe approach. Designed primarily for mmWave directional channels, as presented, it does not cover second-order statistics analysis, although fading behavior in this range is literally highly dynamic [30]. Among this group of models, only the more general 5G channel model (MG5GCM) supports the four most challenging scenarios—massive MIMO, V2V, HST, and mmWave [31].

A common attribute for all of the mentioned work is the deficiency of the general channel modeling approach, in the sense of second-order statistics evaluation and translation to informational level models, what this work finds as area of contribution. To achieve this goal, derivation of a new channel transition matrix expression for a two-state Markov process, applied in this analysis, is given. To underline, tothe author’s best knowledge, here, for the first time an algorithm and results for wideband LCR and AFD evaluation are provided. 

## 3. Methods Applied

### 3.1. Propagation Deterministic Mechanism

Real-world multipath/multiscatter propagation scenarios for three typical 5G use cases are depicted as in Figure 1: high-speed train (HST), vehicle-to-vehicle (V2V) in an urban environment, and crossroad (V2Vcrossroad) scenarios. First, case (a) comprises transmission from a fixed base station transmitter to a fast moving receiver in the presence of static point scatterers in the surrounding area. In case (b), either the transmitter or receiver is moving with static scatterers between, in the way that they are out of direct line of sight. Under (c), besides the transmitter/receiver, a larger number of scatterers are moving on their way, also. This way, in order to compare, we gradually increased the portion of channel non-stationarity. Under the effect of multipath propagation, the signal at the receiver is formed by a combination of multiple signal components, attenuated and delayed. The basic assumption to build our deterministic model was planar wave propagation of the transmitted and modulated (RF) baseband signal *x_CE_*(*t*), which, in standard notation, is given by expression
(1)$$x_{RF}\left(t\right)=Re\left[x_{CE}\left(t\right)e^{j2\pi f_{0}t}\right]$$
and the signal received through multipath scattering surrounding and free-space propagation is
(2)$$y_{RF}\left(t\right)=Re\left\{\overset{N\left(t\right)}{\underset{i=1}{\sum }}c_{i}\left(t\right)e^{j\phi_{n}\left(t\right)}x_{CE}\left(t-\tau_{i}\right)e^{j2\pi f_{0}t-\tau_{i}}\right\}=Re\left[r\left(t\right)e^{j2\pi f_{0}t}\right]$$
where *N*(*t*)is the time-varying number of scatterers, $c_{i}\left(t\right)e^{j\phi_{n}\left(t\right)}$ magnitude and phase contribution from scattereri, $\tau_{i}$ propagation delay of each multipath component, and *f*_0_carrier frequency. Hence, the received complex (baseband) envelope is
(3)$$r\left(t\right)=\overset{N\left(t\right)}{\underset{i=1}{\sum }}c_{i}\left(t\right)e^{j\varphi_{n}\left(t\right)}e^{-j2\pi f_{0}\tau_{i}}x_{CE}\left(t-\tau_{i}\right)$$

For planar wave propagation, at the receiver side, following that τi=dic, with *c* speed of light, $d_{i}$ is *i*^th^ component path length and λ0=cf0 carrier wavelength, we obtain
(4)r(t)=∑i=1N(t)ci(t)ejφi(t)e−j2πλ0dixCE(t−dic)

Mobile user equipment is moving with some constant velocity *V*, causing the existence of Doppler frequency at each beam *i* that arrives at the receiver with the angle *α_i_.* At this instance, we focus on fast variations of a signal complex envelope at the receiver (fast fading) and when no direct line-of-sight is available. For analysis of fast fading separately from slow variations (slow fading due to distance change), one must define a small localarea, where observation takes route section Δ*l*of typically 10–40 wavelengths (λ) of transmitted signal, taken from our recent work [32]; see Figure 2a. Samples of r(t) are taken at some fractions of wavelength *F*,at each $\Delta x=\lambda_{0}/F$ point of mobile movement, equivalently each $\lambda_{0}/FV$ second. The contribution with Doppler shift from *i*^th^ ray, in summary the received signal, depends on the angle of arrival $f_{Di}=f_{max}cos\alpha_{i}$, where $f_{max}=V/\lambda_{0}$ presents maximum Doppler shift. We observe changes either in amplitude or phase. Two factors in (4) comprise phase changes:$\varphi_{i}\left(t\right)$, introduced by the scatterer reflection coefficient and here, omitted due to complexity;$\Delta \phi_{i}$, originating from phase rotation along the electrical distance of a wave.
(5)$$\Delta \phi_{i}=\frac{2\pi }{\lambda_{0}}d_{i}$$

It should be noted that realization without term $\varphi_{i}\left(t\right)$ shows comparable results with empirical measured models. Now, we observe amplitude changes of r(t):$c_{i}\left(t\right)$*,* which represent ith scatterer contribution to overall magnitude of received signal;xCE(t−dic), term that follows changes of transmitted signal.

From link budget theory, supposing a small area and isotropic point scatterers of the same radar cross-section, we calculate contributions of each scatterer normalized with respect to direct path $d_{DP}$ and denoting a transmitter–scatterer path with $d_{TSc}$ and scatterer–receiver path with $d_{ScR}$,
(6)$$c_{i}=\frac{d_{DP}}{d_{TSc}}\cdot \frac{1}{d_{ScR}}$$
and hence, denoting overall receiving signal amplitude with $a_{i}$, calculate it according to free-space power decay law
(7)$$\underset{i}{\sum }a_{i}^{2}=10^{P_{r}/10}$$
where *P_r_* is received power attenuated relative to transmission power *P_t_* (link budget calculation).
(8)$$P_{r}=P_{t}-10\cdot log\;c_{i}^{2}$$

This way, the narrowband channel model was built up, leading to deterministic expression of receiving complex envelope term
(9)$$r\left(t\right)=a_{i}e^{-j2\pi \;\frac{V}{\lambda_{0}}\;t}=a_{i}e^{-j\;\frac{2\pi }{\lambda_{0}}d_{i}}$$

Hence, observing the small area from Figure 2a, for single (measurement) point *x*[*n*] of summary *N_samples_* samples, introducing scattering amplitude contribution vector a and distance path (T-Sc-R) vector d[n], the received narrowband signal vector for our deterministic model is constituted
(10)$$r\left[n\right]=\;ae^{-j\;\frac{2\pi }{\lambda_{0}}d\left[n\right]}$$
where $n=1,\ldots ,\;N_{SC}$ and *N_SC_* denotes number of scatterers. Now, by vector concatenation, a suitable matrix for the received complex envelope r of dimensions *N_SC_* × *N_samples_* is composed for the entire small area. In Figure 2b and further, the magnitude is normalized with reference to root mean square (RMS) value.

### 3.2. Level Crossing Rate and Average Fade Duration

Before going onward, it is desirable to illustrate subject metrics, as they show how often a complex envelope crosses the defined threshold and how long, on average, the signal remains below that level (Figure 2b).For comparison with our deterministic approach, let us briefly recall some basic analytics. Rate that complex envelope *r* is crossing level *L*, it is derived from its definition:(11)$$LCR_{L}=\overset{\infty }{\underset{0}{\int }}\dot{r}p_{r\dot{r}}(L,\;\dot{r})\;d\dot{r},\;r\geq 0$$
where $p_{r\dot{r}}\left(L,\dot{r}\right)$ denotes joint probability density function of stochastic variable r(t) particularized at level *L* and its derivative r ˙(t)=drdt at the same time instant. By definition, AFD is the expected value of the length of the time intervals in which the stochastic process is below a given signal level *L* and is calculated by
(12)$$AFD=\frac{CDF(r<L)}{LCR_{L}}$$
and CDF(r<L) is cumulative distributive function probability that stochastic process (complex envelope *r*) is less or equal to *L*. These metrics, although often hard to express analytically, numerically are not intensive to calculate and measure. The urban environment, on the other hand, is often statistically described with Rice or Rayleigh distribution, where it is shown that
(13)$$LCR_{L}=\sqrt{\frac{\beta }{2\pi }}\frac{L}{\sigma_{0}^{2}}e^{-\frac{L^{2}}{2\sigma_{0}^{2}}}=\sqrt{\frac{\beta }{2\pi }}R_{r}\left(L\right),\;L\geq 0,\;$$

Showing that practically, LCR depends on autocorrelation function $R_{r}\left(L\right)$, where $\sigma_{0}^{2}$ denotes mean power and *β* is short notation of negative curvature for the autocorrelation function at the origin*,*
(14)$$\beta =-{\frac{d^{2}}{d^{2}\tau }R\left(\tau \right)|}_{\tau =0}$$

At the same time, the autocorrelation function is found to be
(15)$$R\left(\Delta x\right)=J_{0}\left(\frac{2\pi }{\lambda }\Delta x\right)$$
with *J*_0_ noted the well-known Bessel function of first kind and zero order. Note that distance and time are connected here through Δx=VΔt=Vτ. Analogously, for AFDin the Rayleigh channel, we can express
(16)$$AFD_{Rayleigh}=\sqrt{\frac{2\pi }{\beta }}\frac{\sigma_{0}^{2}}{L}\left(e^{-\frac{L^{2}}{2\sigma_{0}^{2}}}-1\right),\;L\geq 0,\;$$

Although it is often hard to analytically express for different statistical distributions, LCR and AFD are not intensive to calculate and measure. 

For given r, the first thing is to configure a deterministic LCR vector. This is done by taking integer values of signal levels from lowest to highest r. Then, for the single integer threshold level L, count the number of crossings (samples) across vector r. Using the same levels, we calculate AFD vector across r by counting the number of samples under the threshold and dividing with number of fading intervals.

In order to evaluate wideband values, here is proposed an extension over the arbitrary frequency band where, for single frequency, LCR and AFD are calculated. Steps related to the appropriate matrices and vectors calculation are given in Table 1.

Relating the Table 1 parameters to the 5G architecture, let us observe the physical resource block (PRB), representing the basic scheduling unit, comprised of 12 subcarriers over one orthogonal frequency-division multiplexing (OFDM) symbol, spaced apart by Δf = 2^μ^× 15 KHz, where μ∈{0, 1, 2, 3, 4}, denotes the applied 5G transmission scheme or, so called, numerology. Therefore, for our **f**_axis_ parameter, let us use frequency range starting from 3500 MHz with the step aligned according to Δf. Another view is from the PRB granularity level, where we could set the resolution to 12∙Δf. In the time scale, the basic scheduling unit in 5G is indicated by symbol length indicator value (SLIV). SLIV is practically transmission time interval (TTI in LTE terminology), consisting of a variable number of OFDM symbols (minimum 2) and chased to be aligned with a small area and duration of N_samples_∙t_s_, where t_s_ is the time step that we take our samples and equals t_s_ = Δx/V. Determining optimal SLIV (TTI) and selecting the most efficient PRBs enhances the wideband approach to use for adaptive transport block size (TBS) calculation, presumable in 5G NR [33].

### 3.3. Finite State Markov Chain Constitution

#### 3.3.1. State Classification Model

One application worth discussing is the constitution of the FSMC model, as in [34]. In this work, normalized signal level is used to be the channel state classification benchmark, as depicted in Figure 3a. 

This means that for appropriate signal level range, a two-state Markov model can be modeled (Figure 3b). This useful model can further be used to predict either PER, throughput, etc. Suppose the presence of a mobile user in a network which, according to the measured value of signal level, can be classified in one of *k* specified signal level regions, assumed to have the same traffic properties and hence, same QoS requirements. Physically, it might represent classes of users with proximate velocities, device classes, users inside similar traffic densities, characterized with some guaranteed QoS values inside *k*^th^ region. This approach is usually used for adaptive modulation and coding (AMC) algorithms employed by higher network layers. We suggested here one realization, defining two possible states: the signal belongs to (varies in limits of) class k, and signal does not belong to class k, as depicted in Figure 3b. These states are described with steady state probabilities “in state k”, $p_{k}$, “out of state k”, $p_{o}$, and transition probabilities $p_{ko}$, $p_{ok}$. The channel state transition matrix configured this way is
(17)$$P_{Ch}=\left|\begin{array}{cc} p_{k} & p_{ko}\\ p_{ok} & p_{o} \end{array}\right|$$

#### 3.3.2. Channel Matrix Derivation

Let us associate an AMC scheme ω to the mobile user if its current ***r*** falls in region k, $r\in (r_{\omega -1}^{b},r_{\omega }^{b}]$, $r_{\omega }^{b}$ denoting the boundary signal level for ∀ω, used to select AMC scheme. The channel assumption is that transitions are smooth or made only in neighboring regions,
(18)$$p_{k,k^{\prime }}=0{\;\mathrm{and}\;p}_{k,k+1}=0,\;\forall \left|k-k^{\prime }\right|>1\;$$

To determine transition probability to “out of state k”, we perform simply
(19)$$p_{ko}=p_{k,k-1}+p_{k,k+1}-j_{k,k-1,k+1}$$
where $j_{k,k-1,k+1}$ is the probability that both transitions happened at one time. Assuming that for one user, signal can transit only to one state in the same time instant, we have $j_{k,k-1,k+1}=0$. Following steps from [35], denote cumulative density function of state *k* with $CDF_{k}$ and set
(20)$$p_{k,k+1}=\frac{LCR_{k+1}\cdot \tau }{CDF_{k}}$$
(21)$$p_{k,k-1}=\frac{LCR_{k}\cdot \tau }{CDF_{k}}$$

We now derive our probabilities by substituting (20) and (21) in (19),
(22)$$p_{ko}=\frac{{LCR}_{k+1}\cdot \tau }{{CDF}_{k}}\;+\;\frac{{LCR}_{k}\cdot \tau }{{CDF}_{k}}=\frac{({LCR}_{k+1}+{LCR}_{k})\cdot \tau }{{CDF}_{k}}$$

As rows in the (10) summary give a probability of 1, then
(23)$$p_{k}=1-\frac{({LCR}_{k+1}+{LCR}_{k})\cdot \tau }{{CDF}_{k}}$$

In the same way,
(24)$$p_{\mathrm{ok}}=p_{k-1,k}+p_{k+1,k}-j_{k-1,k+1,k}=p_{k-1,k}+p_{k+1,k}$$
(25)$$p_{\mathrm{ok}}=\frac{{LCR}_{k}\cdot \tau }{{CDF}_{k-1}}\;+\;\frac{{LCR}_{k+1}\cdot \tau }{{CDF}_{k+1}}$$
(26)$$p_{o}=1-\left(\frac{{LCR}_{k}}{{CDF}_{k-1}}\;+\;\frac{{LCR}_{k+1}}{{CDF}_{k+1}}\right)\;\cdot \tau \;$$

Here, we shortly look back to parameters τ and $CDF_{k}$. First, this presents a transmission time interval (fixed 1ms in LTE) or multiple OFDM symbols (SLIV in 5G NR), as mentioned, and within channel response remains invariant. Channel response may change from interval to interval, as assumed. Signals in 5G are transmitted in bursts, so we can take this time as the duration of one burst. Another viewpoint is to consider it as packet transmission time, where τ=Packet_length/Symbol_rate. The second parameter is one of first-order statistics expressed as
(27)$$CDF_{k}=\overset{L_{k+1}}{\underset{L_{k}}{\int }}prob\left(r\right)dr$$

After completing all the elements, we conclude our observation, leaving space for further exploitation of the presented concept for QoS metrics evaluation.

## 4. Symbolic Encapsulation Roadmap

Thanks to the concept of microsimulation semi-symbolic analysis [36], all variables and operations are encapsulated in one framework, which is accessed by simply calling its reference. The encapsulation process is the store of all relevant parameters, data, and behavior in one component—a symbolic encapsulation point. The complete procedure is shown in Figure 4. The figure represents a roadmap from inputs to symbolic and numerical outputs. Symbolic outputs represent expressions in closed-form, while numerical outputs are values for generating visual characteristics. The entrance of the roadmap is input signal ⓐ that can be measured or simulated. In Section 3, the way to define the parameters of second-order statistics ⓑ is shown. The next phase ⓒ involves complex calculations in order to bring the parameters of the second-order statistics (LCR and AFD) into the form for fast computation [37,38,39,40]. Microsimulation semi-symbolic elements ⓓ are then applied to obtain closed-form expressions ⓔ that provide encapsulated frames ⓕ. They can be further simplified ⓖ if possible or manipulated ⓗ. At the same time, numerical values can be obtained ⓘ and use them to obtain the characteristics ⓙ. If the input signal is simulated, in fact computer-generated, these parameters can be set via ⓚ. In this framework, it is possible to set the speed of the mobile users, the number of scatterers, fast Fourier transform (FFT) size, simulation length, sampling rate, frequency free space loss, base station distance, and radius of scatterers. In [37], the methodology of solving very complex scenarios in wireless communications is presented. The results of the error function of symbol error probability in terms of the number of the iteration are illustrated. Paper [38] presents, in detail, the methodology of the key parameters that simplify complex calculations. The paper [39] shows how it is possible to speed up a complex calculation where the relative errors do not exceed more than 8% between the original and the approximate value. In [36], the calculation speed was taken into account, which speeds up the calculation of outage probability by 955 times, and reduces the number of mathematical operations by almost 4 times, while for statistical parameters of the second-order, LCR is accelerated by 20 times and AFD by 15 × 10^3^ times, for a relative error ranging between 0.5% and 2.5%.

Figure 5 shows a full graphical environment for monitoring all parameters. Figure 5a shows the magnitudes of the complex envelope in time (s) and dB as well as the complex envelope normalized to RMS. Figure 5b shows modulo π of the complex envelope, absolute phase of the complex envelope, and phase histogram. Figure 5c provides the autocorrelation function and cumulative distribution function. Figure 5d shows a normalized RF spectrum and a normalized baseband spectrum. Figure 5e shows the LCR and AFD parameters. The visual environment provides the display of 3D features and in the next session, complex examples for the three scenarios are presented.

## 5. Numerical and Simulation Results

### 5.1. HST Scenario

The scenario in which a communication link between a fixed base station and a fast-moving receiver is emulating the transition of a high-speed train, moving up to 500 km/h and causing channel conditions to change rapidly. In order to do our simulation setup, we choose a lower 5G frequency range (up to 7125 MHz), starting from a reference point of 3500 MHz. Looking at (10), computation of the appropriate distance matrix (BS-Scatterer-MS, for each sample point) is done. For matter of convenience, smaller bandwidths then available are shown in Figure 6, intending to clearly visualize regions with the largest number of lost bits. The setup parameters are summarized in Table 2. LCR on the left side and AFD on the right symbolically give information on channel quality across specified bands. Higher LCR peaks point on band parts where signal is more fluctuating; therefore, in our transmission scheme selection, we can choose not to use these PRBs, or use them in another scheme (e.g., lower throughput). Regions with longer and shorter duration of fading intervals bring some conclusions about packet loss—Figure 6b for case A, Figure 6d for case B, and Figure 6f for case C. 

Using the provided color map, we can perceive that the largest number of lost bits in this scenario is expected in yellow regions (where over 30 wavelengths reside bellow some level), 3500–3500.25 MHz (250 KHz), next about 250/15~16 carriers and 3502.75–3503.5 MHz (750 KHz), or about 750/15~48 carriers; in all, around 64 carriers. Numerically, evaluation results become available for adaptive PRB aggregation (interleaving) to achieve better transmission. The next trial was with second numerology, case B. From Figure 6b, we see more of the spectrum and new part with high AFD, from about 3505–3506 MHz (1 MHz width). Here, we use 20 KHz frequency resolution, as it was enough with respect to carrier distance (30 KHz). In the third trial, for case C, observation band was expanded to 14.4 MHz (Figure 6c). We doubled scanning resolution, as in this case, subcarrier spacing is 60 KHz. It is obvious about the new part shown with long fades and part from 3511 to 3512 MHz (1 MHz) that is considerably free of fades and can be used for higher data rates.

### 5.2. V2V Scenario

In this scenario, besides the moving receiver, the moving transmitter is included, also. Therefore, for each sample point at the receiver, we dislocate the transmitter for the length it travels with some constant velocity, different from the velocity of the receiver, and calculate distance matrix and hence, the complex envelope, again. For the sake of simplicity, the assumption is that both stations are moving in the same, or opposite, but parallel direction and we use the same absolute values of velocities. No line-of-sight (LOS) is available, so only scattering rays are present, either from roof tops, groups of trees, or, rarely, between buildings (Figure 1b).

This time, time-varying frequency response was generated in order to visualize amount of non-stationarity, conventionally. It is noticeable the difference of number of crossings in this scenario depending on the directions of velocity vector, as shown in Figure 7a,b. When moving in the same direction, relative speed is lower, and so it is the influence of Doppler frequencies. Variations of channel become more degrading when moving in opposite, resulting in fades deeper than 50 dB during 100 ms period, compared to same direction movement, Figure 7e,f. Observing these cases across scanned frequency bands, the signal differently stays under the fading area (Figure 7c,d), suggesting the creation of virtual resource blocks, by adequate mapping of good PRBs in one continual resource. AFDs, in some regions, are almost ten times longer than others (10^4^ vs. 10^5^ wavelengths) and consequently assumed to have incremented PER, for example. Other simulations stay further available to try, e.g., by different velocities or incorporating arbitrary moving directions, which is outside of the present observation. 

### 5.3. V2Vcrossroad Scenario

For the third scenario observed, moving scatterers are included (more channel variations). As mentioned, this use case can be compared to heavy crossroads with congested traffic. Now, for every sample time unit, either the transmitter, scatterers (all or some), or receiver change their position when acquiring a complex envelope data matrix. To make more non-uniform changes in scattering position, we set different velocities and directions as vehicles moving in areal situation. 

As in the first scenario, three cases are simulated, Figure 8. Each case considers non-LOS existence. Notice peaks in the AFD diagram saying what carriers offer better performance. Generally, looking with higher resolution gives better cognition about passband usage. All LCR and AFD results could be stored and presented in a table for analysis.

## 6. Conclusions

In this work, usage of second-order statistics, namely LCR and AFD, as metadata to QoS modeling were emphasized. By using complex envelope signal samples collected from a small travel route as input, we achieved the generating of enough statistics to describe the non-stationary nature for different types of mobile channel, showing generality of approach. Present general channel 5G models can benefit from this approach by extending their missing features for dual-mobility or high-mobility cases. With the extension to wideband case, it is more clear how observed mobility affects different bandwidth parts, which optimization engineers can utilize for adaptive modulation and coding schemes, resource blocks interpolating, optimal buffering, transport block size calculating, and other mechanisms for gaining targeted throughput, packet error rate, latency, etc. Using the presented 5G tool roadmap, we simplified complexity of calculations, reducing simulation time with an acceptable amount of accuracy. By means of microsimulations and their encapsulation through symbolic representation, symbolic encapsulation point manifested as a promising tool for a kind of cross-layer network design and optimization.

Directions of further investigations are seen in integration of the symbolic encapsulation point tool into existing general channel 5G models with sophisticated propagation mechanisms (map-based) and applied 5G technologies to provide more accurate data and confirmation by measurements. Another option for work is extension to symbolic encapsulation of Markov chains and that way, through cross-layer modeling, an approach to wanted quality of service parameters evaluation in a dynamically changing wireless environment. This implicates reduced time for network planning and optimizing, a smaller number of (real-time) fading measurements, and field mobility testing, often connected with high costs and ability to evaluate user-experience data in a dynamic channel environment.

## Figures and Tables

**Figure 1 entropy-22-01151-f001:**
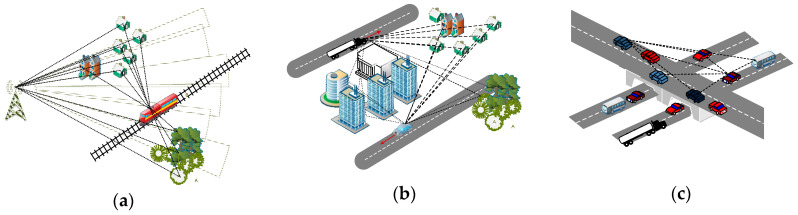
Examples of typical 5G propagation scenarios: (**a**) high-speed train; (**b**) urban vehicle-to-vehicle; (**c**) crossroad.

**Figure 2 entropy-22-01151-f002:**
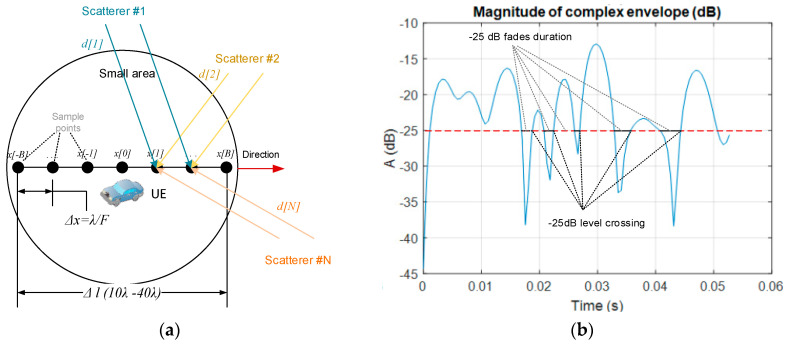
Description of small-scale fading analysis: (**a**) small area; (**b**) level-crossing rate (LCR) and average fade duration (AFD) definition.

**Figure 3 entropy-22-01151-f003:**
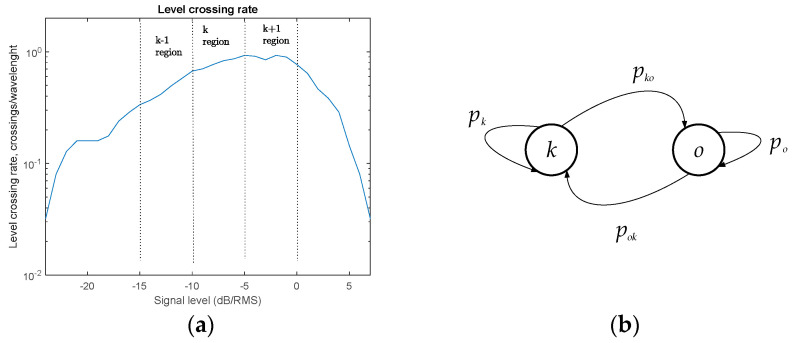
State classification model: (**a**) LCR classification; (**b**) Two-state Markov model.

**Figure 4 entropy-22-01151-f004:**
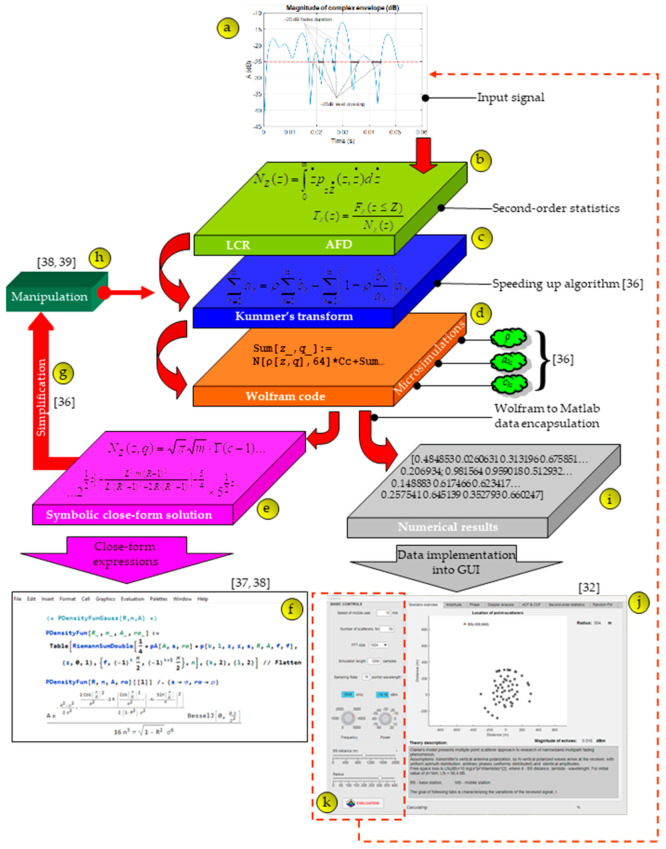
5G tool roadmap.

**Figure 5 entropy-22-01151-f005:**
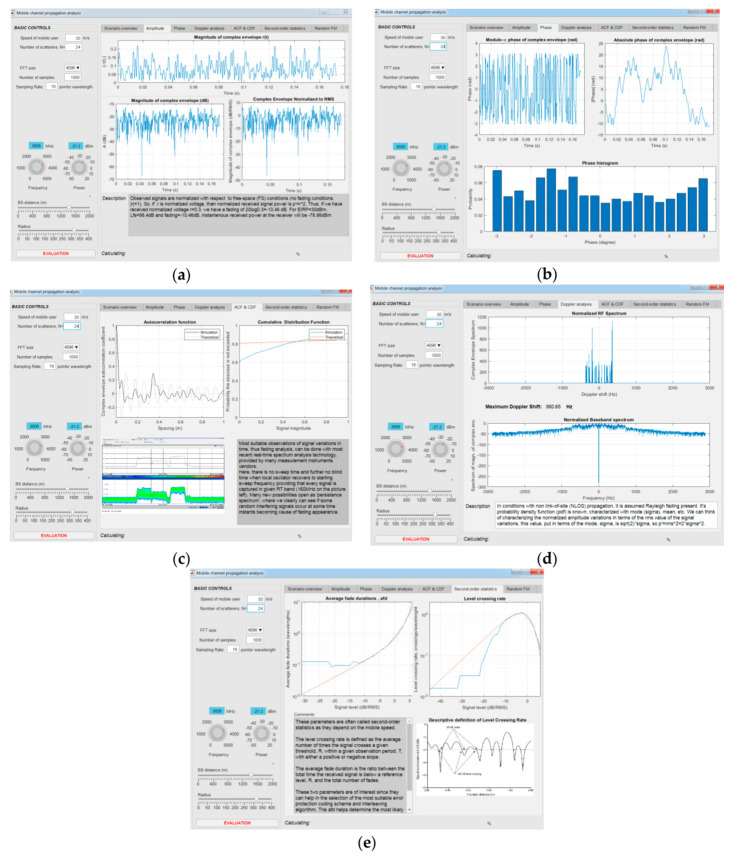
Full visual environment of tool: (**a**) magnitude; (**b**) phase; (**c**) autocorrelation function (ACF) and cumulative distribution functions (CDF); (**d**) Doppler analysis; (**e**) level-crossing rate (LCR) and average fade duration (AFD).

**Figure 6 entropy-22-01151-f006:**
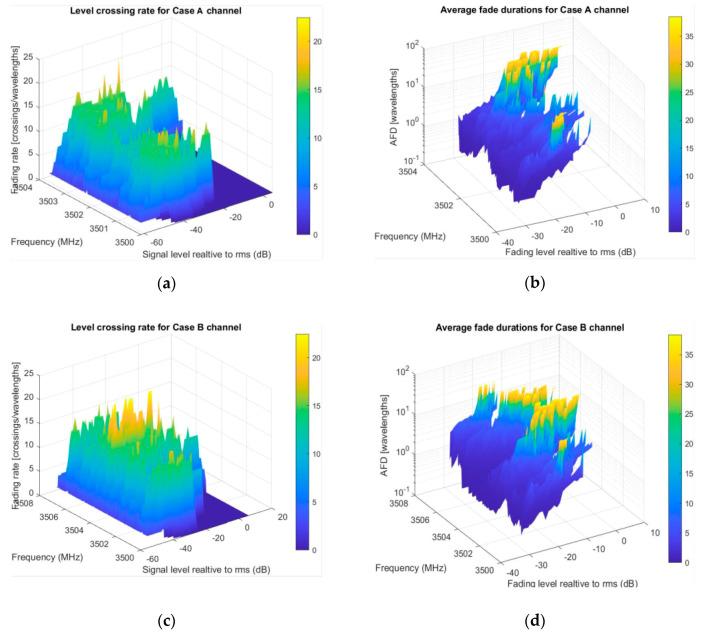
Evaluation results for second-order statistics in high-speed train (HST) scenario: (**a**) Case A passband LCR; (**b**) Case A passband AFD; (**c**) Case B passband LCR; (**d**) Case B passband AFD; (**e**) Case C passband LCR; (**f**) Case C passband AFD.

**Figure 7 entropy-22-01151-f007:**
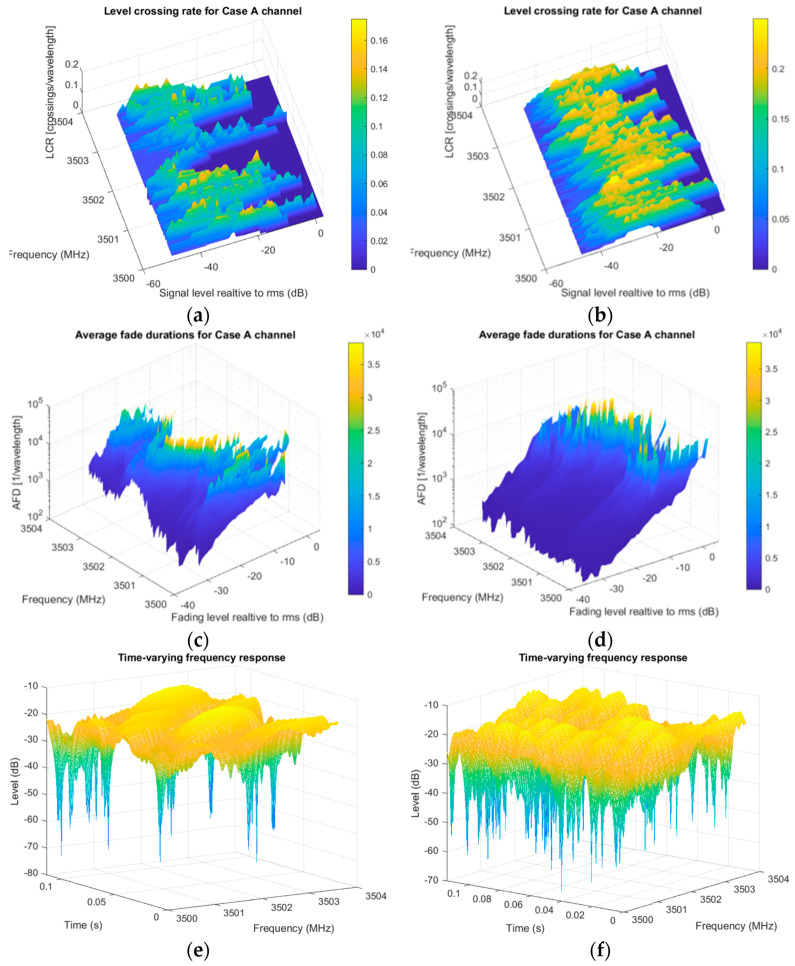
Evaluation results for second-order statistics in V2V scenario: top row shows same direction (**a**) passband LCR; (**b**) passband AFD; (**c**) time-varying frequency response; bottom row shows opposite direction; (**d**) passband LCR; (**e**) passband AFD; (**f**) time-varying frequency response.

**Figure 8 entropy-22-01151-f008:**
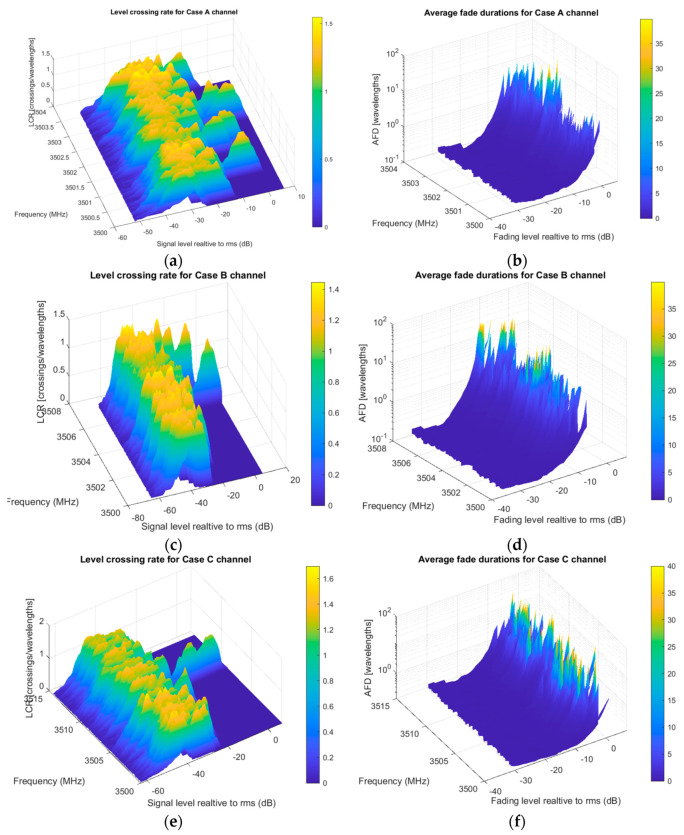
Evaluation results for second-order statistics in V2V crossroad scenario: (**a**) Case A passband LCR; (**b**) Case B passband AFD; (**c**) Case B passband LCR; (**d**) Case B passband LCR; (**e**) Case C passband LCR; (**f**) Case C passband LCR.

**Table 1 entropy-22-01151-t001:** Steps of wideband extension for evaluating LCRand AFD.

Step	Pseudocode
1	*Input N_samples_, N_SC_*, *Δf, t_s_, P_t_*
2	*B=* (*N_samples_*−1)/2;
3	*Generate distance matrix **D**=* | d[−B]…d[0]…d[B]|
4	*Calculate **c_i_** according to (6) for every of N_SC_ scatterers*
5	*Calculate **P_r_** using (8)*
6	*Calculate scatterers magnitude vector* **a** *according to (7)*
8	***f****_axis_* = (start: *Δf*: end)
9	**W** =zeros (*N_samples_*, length(***f****_axis_*))
10	**LCR, AFD =** zeros (length(***f****_axis_*), *N_samples_*)
11	**For***n*=1 **to***N_samples_*
12	**For** b=1 **to** length(***f****_axis_*)
13	**For** s=1 **to** *N_SC_*
14	Λ =c/***f****_axis_*(b)
15	**W**(n,b)= **W**(n,b) + a(s)×exp(-j×2π/λ)× ***D***(*s*,*n*)
16	**end**
17	**end**
18	**end**
19	**For** b=1 **to** length(***f****_axis_*)
20	**r** = transpose(column(**W**,b))
21	*Calculate root mean square* RMS *of* **r**
22	*Normalize as* abs(**r**)/RMS
23	*Calculate level crossing rate* **LCR** *and average fade duration* **AFD** *across* **r**
24	*Update matrices* **LCR**(p,1:length(**LCR**)) *and* **AFD**(p,1:length(**AFD**))
25	**end**

**Table 2 entropy-22-01151-t002:** Simulation setup parameters.

Numerology	Case A	Case B	Case C
SCS (kHz)	15	30	60
Bandwidth (MHz)	3.6	7.2	14.4
Resolution (kHz)	180	360	720
Symbol duration (μs)	66.7	33.3	16.7
Simulation time (4 symbols)	200	100	50

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
