# Peer review of "A Symbolic Encapsulation Point as Tool for 5G Wideband Channel Cross-Layer Modeling"

_entropy, 2020, doi:10.3390/e22101151_

Round 1
Reviewer 1 Report
Dear author(s),
I read your paper with great interest and enthusiasm. This is an interesting paper in 5G channel modelling. The scientific contribution is quite good. However, the paper requires a good revision to improve English language/readability. The following issues need to be looked at for further improvement of the paper.
- Abstract - It is not well justified why this research is needed. It is unclear from this paper/abstract what's the original contribution to the field is. This needs to be clearly stated.
- Introduction: This is written in a superficial manner and it requires improvement.
- Related work: You need to provide an in-depth literature review to find the gap (areas for contribution).
- Conclusion: The conclusion is bit weak. It requires improvement. State clearly what has been achieved in this paper and plan for future research directions.
- Simulation results: The system simulation part of the paper is bit weak. It is not clear how the simulation model was developed. The simulation model validation and accuracy of the simulation results have not been addressed at all.
The following questions need to be addressed.
What simulator did you use?
What's the accuracy of the simulation results (in terms of confidence interval and statistical error)?
What's the length of the simulation?
How did you validate simulation models/results? - Implications: The practical system implications is missing in this paper. Discuss the system implications - how practitioners can benefit from this research. This can be discussed in the concluding sections.
Author Response
The authors are gratefull for the comments of the reviewers that enormously helped quality improvement of the work and this notification will be added in the acknowledgement. As requested, comments of reviewers are provided in received order and carefully answered to the best of authors knowledge with clear notification (remarked as yellow) what has been included in the edited manuscript:
Reviewer I Comments:
The following issues need to be looked at for further improvement of the paper.
- Abstract- It is not well justified why this research is needed. It is unclear from this paper/abstract what’s the original contribution to the field is. This needs to be clearly stated.
Answer: In order to improve the perception of the paper’s contribution, the abstract is rephrased and stated as follows. Here, we underline-indicate main missing element of the field, and contribution is clearly enumerated:
Considering that networks based on New Radio (NR) technology are oriented to provide services of the desired quality, it becomes questionable how to model and predict targeted QoS values, specially if the physical channel is dynamically changing. In order to overcome mobility issues, we aim to support the evaluation of second-order statistics, namely level-crossing rate (LCR) and average fade duration (AFD) that is missing in general channel 5G models. Presenting results from our symbolic encapsulation point 5G (SEP5G) additional tool, we fill this gap and motivate further extensions on current general channel 5G models in this direction. As a matter of contribution, we clearly propose (i) additional tool for encapsulating different mobile 5G modeling approaches (ii) extended, wideband, LCR, and AFD evaluation for optimal radio resource allocation modeling, and (iii) lowering computational complexity and simulation time regarding to analytical expression simulations in related scenario-specific 5G channel models. Using our deterministic channel model for selected scenarios and comparing it with stochastic models, we show steps towards higher-level finite-state Markov chain (FSMC) modeling, where mentioned QoS parameters become more feasible, placing symbolic encapsulation at the center of the cross-layer design.
- Introduction: This is written in superficial manner and it requires improvement.
Answer: For the introduction improvement we conducted following changes:
In row 53 instead of word actual, we placed „representative general“ and after word channel placed „5G“, to unambiguously clarify what kind of approach/group of channel models available this contribution is dedicated to.
Further, from row 56 to 64 we added „This conclusion comes from the fact that only one of top-ten general channel models, according to acknowledged reference, is able to support both dual-mobility and high-mobility, as two of four most challenging scenarios in 5G. On the other hand, observations made in scenario-specific 5G channel models of mobile channels with LCR and AFD evaluation, however, do not provide generality in terms of supporting various 5G technologies and adapting to different scenarios, as general channel 5G models do. Second, all observations are made in narrowband sense, limitating perception for bandpart resourse schedulling. Third, analytical expressions for LCR and AFD are often complex, comprising integral functions that diverge [8] and consequently computationaly intensive.“
As final improvement of introdution, we changed consequent rows to „Assuming some typical 5G propagation scenarios, figure 1, as for contributions of this work we bring: (i) additional tool for encapsulating different mobile general channel 5G modeling approaches (ii) wideband extension of LCR and AFD evaluation for optimal radio resource allocation modeling and (iii) lower computational complexity and simulation time reduction related to scenario-specific 5G channel models“.
- Related work: You need to provide in-depth literature review to find the gap (areas of contribution)
Answer: To the extent of related work section, we:
Replaced word contrubution with „applications“ in row 102 and accordingly rephrased next two rows.
Changed rows 110-119 stating „Channel modeling for 5G evolved during past few years. Much effort was shown in survey [17] to aggregate related topic. Related authors differ three approaches to the field: channel measurements, scenario-specific 5G channel modeling and general 5G models. Only few measurements and scenario-specific channel models tackled LCR and AFD in order to characterize various aspects of the dynamic behaviour of envelope fluctuations in [18], and use it for higher-layer (FSMC) modeling scenarios [19], [20] . Although in [19] wideband was observed, neither of them included these metrics in wideband aspect. In [21], validity of FSMC approach was confirmed by field measurements in case specific, HST scenario, but no work has found for general 5G models. More further, complexity of analytical expressions derived is these sceanrio-specific models is often numericaly intensive. From general channel 5G modeling view...“
Added sentence, row 138-140, „Among this group of models, only more general 5G channel model MG5GCM supports the four most challenging scenarios, massive MIMO, V2V, HST and mmWave [31].“
This way, we introduced 6 references more (including one inserted in introductory, see (2)) and complemented in-depth analysis to show connection of current gaps in the field and stated contributions in the paper.
At the end of the section, we replaced: word integral with „general“ in row 141, phrase –is trying to contribute for with „finds as area of contribution“ in row 143 and word contributed with „given“ in row 144.
- Conclusion: The conclusion is bit weak. It requires improvement. State clearly what has been achieved in this paper and plan for future research directions.
Answer: Considering the comment, conclusion section sentences are put in more consistent shape and state (rows 381-402):
In this work, usage of second-order statistics, namely LCR and AFD, as metadata to QoS modeling were emphasized. By using complex envelope signal samples collected from a small travell route as input, we achieved to generate enough statistics to describe non-stationary nature for different types of mobile channel, showing generality of approach. Present general channel 5G models can benefit from this approach by extending its missing features for dual-mobility or high-mobility cases. With the extension to wideband case, it is more clear how observed mobility affects on different bandwidth parts, which optimization engineers can utilize for adaptive modulation and coding schemes, resource blocks interpolating, optimal buffering, transport block size calculating and other mechanisms for gaining targeted throughput, packet error rate, latency, etc. Using presented 5G tool roadmap, we simplified complexity of calculations, reducing simulation time with acceptable amount of accuracy. By means of microsimulations and their encapsulation through symbolic representation, Symbolic encapsulation point manifested as promising tool for a kind of cross-layer network design and optimization.
Directions for further investigations are seen in integration of the symbolic encapsulation point tool into existing general channel 5G models with sophisticated propagation mechanisms (map based) and applied 5G technologies to provide more accurate data and confirmaton by measurements. Another option for work is extension to symbolic encapsulation of Markov chains and that way, through cross-layer modeling, approach to wanted quality of service parameters evaluation in dynamicaly changing wireless environment.
- Simulation results: The system simulation part of the paper is bit weak. It is not clear how the simulation model was developed. The simulation model validation and accuracy of the simulation results have not been addressed at all.
Answer: Thanks to reviewer for these observations.
The 5G tool is developed by integrating previously developed microsimulation-semi-symbolic analysis and has been tested in three different scenarios described in the work in sections 5.1, 5.2, and 5.3. The initial values of the simulations are given in Table 2. The results were obtained in a few seconds using all the implemented algorithms for speeded up computations.
- The following questions need to be addressed.
- What simulator did you use?
- What's the accuracy of the simulation results (in terms of confidence interval and statistical error)?
- What's the length of the simulation?
- How did you validate simulation models/results?
Answer: Thanks to the reviewer about this observation. We would ask the reviewer to accept our unified answer to the questions asked.
Namely, the simulator is developed in several segments. All segments were analyzed, tested, and verified, and all performances were presented and published in previous papers [35], [36], [37], [38], separately. We thought it was not expedient to repeat all the analyzes and proofs, but we involved all the previous algorithms, and that would not be the focus of this paper. All previous analyzes, algorithms, and methods are now integrated into one point, more precisely encapsulated. Since these are symbolic tools, it is necessary to implement a numerical tool with a graphical environment that would ensure that the performance of the observed parameters can be seen, and this is done here. This answer is described in section 4. In [36], the methodology of solving very complex scenarios in wireless communications is presented. The results of the error function of symbol error probability in term of the number of the iteration are illustrated. Paper [37] presents in detail the methodology of key parameters that simplify complex calculations. The paper [38] shows how it is possible to speed up a complex calculation where the relative errors do not exceed more than 8% between the original and the approximate value. In [35], the calculation speed was taken into account, which speeds up the calculation of outage probability by 955 times, and reduces the number of mathematical operations by almost 4 times, while for statistical parameters of the second-order LCR is accelerated by 20 times, while AFD by 15∙103 times for a relative error ranging between 0.5 and 2.5%.
- Implications: The practical system implications is missing in this paper. Discuss the system implications – how practitioners can benefit from this research. This can be discussed in the concluding sections.
Answer:
This implicates reduced time for network planning and optimizing, smaller number of (real-time) fading measurements and field mobility testing, often connected with high costs and ability to evaluate user-expirience data in dynamic channel environment.
This tool will be very useful for researchers who want to get fast results for second-order statistical parameters and their further study, while engineers, especially those involved in 5G mobile network design, will be able to use these tools in real-world environments when planning the site and the position of 5G base stations.

Reviewer 2 Report
The paper is very well written and is a comprehensive study.
In order to increase the quality of the paper, the figures should be improved, for example most of the figures have labels, legend etc are very small and images are of low quality. This needs to be improved.
The paper can be accepted in the current form, with some ammendments to the images.
Author Response
The authors are gratefull for the comments of the reviewers that enormously helped quality improvement of the work and this notification will be added in the acknowledgement. As requested, comments of reviewers are provided in received order and carefully answered to the best of authors knowledge with clear notification (remarked as yellow) what has been included in the edited manuscript:
Reviewer II Comments:
- In order to increase the quality of the paper, the figures should be improved, for example most of the figures have labels, legends etc. are very small and images are of low quality. This needs to be improved. The paper can be accepted in the current form with some ammendments to the images
Answer: Some ammendments to the images are provided as in the text, so as to be more clearly:
Thanks to the reviewer for pointing out the flaws around the pictures. Figures 2, 4, 6, 7 and 8 have been corrected.

Round 2
Reviewer 1 Report
The review has been done with my satisfaction.